# A Review on Reconfigurable Liquid Dielectric Antennas

**DOI:** 10.3390/ma13081863

**Published:** 2020-04-16

**Authors:** Elizaveta Motovilova, Shao Ying Huang

**Affiliations:** 1Engineering Product Development, Singapore University of Technology and Design, Singapore 487372, Singapore; elizaveta@alumni.sutd.edu.sg; 2Department of Surgery, National University of Singapore, Singapore 119077, Singapore

**Keywords:** reconfigurable antennas, frequency tuning, radiation pattern, liquid dielectric, water

## Abstract

The advancements in wireless communication impose a growing range of demands on the antennas performance, requiring multiple functionalities to be present in a single device. To satisfy these different application needs within a limited space, reconfigurable antennas are often used which are able to switch between a number of states, providing multiple functions using a single antenna. Electronic switching components, such as PIN diodes, radio-frequency micromechanical systems (RF-MEMS), and varactors, are typically used to achieve antenna reconfiguration. However, some of these approaches have certain limitations, such as narrow bandwidth, complex biasing circuitry, and high activation voltages. In recent years, an alternative approach using liquid dielectric materials for antenna reconfiguration has drawn significant attention. The intrinsic conformability of liquid dielectric materials allows us to realize antennas with desired reconfigurations with different physical constraints while maintaining high radiation efficiency. The purpose of this review is to summarize different approaches proposed in the literature for the liquid dielectric reconfigurable antennas. It facilitates the understanding of the advantages and limitations of this technology, and it helps to draw general design principals for the development of reconfigurable antennas in this category.

## 1. Introduction

Modern wireless communication applications require an antenna to have multiple functionalities (e.g., beam steering, direction finding, radar, control, and command) within a limited space [1,2]. Reconfigurable antennas became a popular approach since they can switch between functions within a single structure without resorting to multiple antennas. Antenna reconfiguration is achieved through a deliberate change in the antenna’s geometry and/or electrical behaviour which results in a change in the antenna’s functionalities [3]. Such antennas typically have two or more discretely or continuously switchable states. These different states are normally obtained by changing current paths of the antenna through either rearranging the antenna itself or altering its surrounding medium. Reconfigurable antennas have been widely applied in many modern radiofrequency (RF) systems used for wireless and satellite communication, imaging and sensing [4].

Since the first patent on reconfigurable antennas appeared in 1983 by Schaubert [5], the topic has gained a lot of attention. In the literature, different designs have been proposed to achieve reconfigurability in terms of either frequency, radiation pattern, polarization, or a combination of two or three of the former ones. The desired reconfigurability can be obtained by various reconfiguration techniques that can be incorporated into an antenna design in order to redistribute its surface current through the change in the feeding network, the physical structure of the antenna, or the radiating edges [4].

The reconfiguration techniques can be divided into several categories as shown in Figure 1. Four major types of reconfiguration techniques are used, namely the electrical, optical, physical, and material reconfigurations. Electrically reconfigurable antennas use RF microelectromechanical systems (MEMS), PIN diodes, or varactors to redirect the surface currents. Those that rely on a photoconductive switching elements belong to optically reconfigurable antennas. For physical reconfiguration technique, the geometry of the antenna can be altered with the help of mechanical deformation. Finally, with the help of smart materials such as ferrites, ferroelectrics, liquid metal, liquid crystal, and liquid dielectrics, the reconfiguration of an antenna can be achieved [4,6,7,8,9].

The most widely used reconfiguration technique is the electrical one. RF-MEMS rely on mechanical movement of microscaled switches, offer a good isolation and require minimal power consumption. PIN diodes operate in two modes (ON and OFF) and offer a faster switching speed (1–100 nsec) compared to the MEMS (1–200 μsec). Other designs use varactor, as they can offer a variable capacitance and thus a continuous tuning ability (RF-MEMS and PIN diodes provide only discrete tuning). Even though these switches can be easily integrated into antenna structure, they require high voltage (RF-MEMS, varactors) and biasing lines that may add losses and distort the antenna radiation pattern [4]. An example of an electrical reconfiguration technique applied for WLAN and Bluetooth can be found in Reference [10].

Optical switches use light of a laser diode to activate photoconductive material. These switches do not require any biasing lines and thus eliminate the problem of introducing unwanted interference [4]. However, optical switches are bulky and the integration of the these components into a compact antenna might become a challenge [1].

Physical switches do not require biasing lines, laser diodes, or optical fibers. Moreover, they can potentially offer high power handling capability, lower losses, and allow for continuous tuning [6]. However, their response time is relatively slow and the integration of the reconfiguration element into the antenna structure can be complicated [4].

As for the reconfiguration techniques based on smart materials, liquid crystals, ferrites, and ferroelectrics can be used to reconfigure the substrate of antenna. The change in a material is achieved by changing its permittivity or permiability under different voltage levels (liquid crystals) or a static applied electric/magnetic field (ferroelctrics/ferrites). Liquid crystals are widely used at optical frequencies, however their property of changing the dielectric constant can be used for reconfigurable antennas at microwave frequencies as well [6]. Their power consumption is relatively low, however, they have to be kept at the temperature range between 20 °C to 35 °C in order to stay in the liquid crystal state [6]. The major advantage of the ferrite-based reconfigurable antennas is their high permittivity ($\epsilon_{r}>10$) and permeability (μ>1000) values that allow for antenna miniaturization and wide tuning range. However, the main disadvantage is the relatively complex biasing network required to achieve the maximum tuning range in bulk material and higher DC power consumption [6]. Compared to ferrites, ferroelectric thin films require less bias voltage and have smaller losses [6].

Liquid metal-based reconfigurable antennas are widely used for flexible, stretchable, and wearable electronics and sensors. They offer intrinsic reconfigurability and conformability, as well as high electrical conductivity and low loss. However, they have a relatively low switching speed, limited power handling, and repeatability issues [9]. The research in liquid metal application for reconfigurable antenna is relatively new, however several review papers and books have been already written [9,11,12,13].

Liquid dielectric antennas have drawn a significant amount of attention for a number of reasons, (1) conformability: any antenna shape can be achieved due to the nature of liquid; (2) reconfigurability (both physical and chemical): it is easy to change the resonance frequency and bandwidth by changing the height/width of the liquid stream and the chemical composition; (3) low cost: liquid dielectrics are cheap and easily available compared to more costly liquid metals (e.g., mercury (Hg) or eutectic gallium indium allow (EGaIn)), (4) transparent and biocompatible, (5) high permittivity which helps to miniaturize antennas. Besides antennas, liquid dielectric materials have been applied in the field of metamaterials and metasurfaces [14,15,16,17], for the design of reconfigurable frequency selective surfaces (FSS) [18], absorbers [19,20,21,22,23], sensors [24], reflect-arrays and array lenses [25], and polarization converters [18]. However, these topics fall beyond the scope of the present review and the details can be found in the review papers [15,26].

There are a few books [1,27] and several review papers published [4,6,14,28,29,30] up to date. They provide comprehensive overview and comparison of modern antenna reconfiguration techniques. However, a systematic review on liquid dielectric reconfigurable antennas is still missing although it is an important promising category. To the best of our knowledge, the only review covering liquid dielectric reconfigurable antennas was done in Reference [31], however it was not comprehensive enough. In the present review we summarize the recent progress on this liquid dielectric reconfigurable antennas. The specific focus of this review paper is to systematise and analyse the research on active antennas that utilize liquid dielectric for radiation reconfiguration (both frequency, pattern, and polarization). It should be noted that even though this review is focused specifically on reconfiguration of active antennas by means of a liquid dielectric(s), the concepts discussed here can be applied to other electromagnetic applications such as sensors, filters, arrays, frequency selective surfaces, and meta- surfaces. The purpose of this paper is to thoroughly review state-of-the-art reconfiguration techniques for antennas in order to establish a general classification for the reconfiguration techniques which helps to systematically analyse this research field and draw conclusions on the research perspectives, gaps, and limitations.

For the review, several parameters were used to evaluate and compare the performance of a frequency reconfigurable antennas. The most obvious one is the frequency tuning range, which can be defined as follows [6]
(1)$$TR=\frac{2(f_{H}-f_{L})}{\left(f_{H}+f_{L}\right)}100\%,$$
where $f_{H}$ and $f_{L}$ correspond to the highest and lowest resonant frequencies of a given antenna, where the resonant frequency is defined as the location of the $|S_{11}|$ minima.

On the other hand, the total spectrum defined in Equation (Equation 2) can be used to asses the total antenna working range [6]
(2)$$TS=\frac{2(f_{max}-f_{min})}{\left(f_{max}+f_{min}\right)}100\%,$$
where $f_{max}$ and $f_{min}$ correspond to the maximum and minimum usable frequencies defined as location of the $|S_{11}|\leq 10$ dB.

Another parameter that can be used to quantify the spread of frequencies is the tuning ratio, as given in Equation (Equation 3) [6].
(3)$$TX_{1}=\frac{f_{H}}{f_{L}},\;TX_{2}=\frac{f_{max}}{f_{min}}.$$

In terms of the tuning type, frequency reconfigurable antennas can provide discrete or continuous tuning mechanism. A discrete tuning mechanism will only provide a fixed number of resonant states, while in the case of a continuous tuning mechanism, the resonant frequency can be of any value within a tuning range TR.

Radiation efficiency is another important parameter to be considered. The radiation efficiency of frequency reconfigurable antennas may vary through the tuning range TR, that is why it is important to provide the radiation efficiency values for the whole TR. Moreover, it is best if the radiation efficiency of a frequency reconfigurable antennas can be compared to that of an equivalent antenna with a fixed operational frequency (not reconfigurable). The radiation efficiency is normally compromised by the reconfigurability technique used.

The last parameter is the physical size of the antenna. In this review, the total size in millimeters (mm) and the longest dimension with respect to the shortest wavelength is calculated where possible for comparison between different designs.

In terms of the classification, generally there are two types of frequency reconfigurable antennas that use liquid dielectrics for reconfiguration. The first type of antennas, called liquid-based antennas in this review, have a liquid dielectric as the main radiating part, for example, liquid-based monopole antennas. Such type of antenna design originates from dielectric resonators [32], where liquid dielectrics are used instead of solid ones. Liquid-based antennas are discussed in Section 2. The second type of liquid dielectric antennas uses dielectric liquid to locally modify the currents of a metal-based antenna, such as for example, patch and slot antenna. This type of antennas is called liquid-assisted antennas in this review. When most of the liquid-based antennas are three-dimensional antennas that typically require large quantities of liquid dielectric material, the liquid-assisted antennas typically have a planar and compact design and require less liquid to switch between different frequency states. Liquid-assisted antennas are discussed in Section 3. Following Section 3, discussions are presented in Section 4. Lastly, conclusions are drawn in Section 5.

## 2. Liquid-Based Antennas

A liquid-based reconfigurable antenna is a type of dielectric resonator antenna (DRA). DRAs have been actively used since the first systematic reviews were conducted by Long, McAllister, and Shen in 1980s [33,34,35]. DRAs became the center of interest for antenna applications due to their high radiation efficiency, compact size, and good match to most commonly used transmission lines [36]. Most of the designs use a slab of dielectric material mounted on a ground plane and excited by an aperture feed or by a probe inserted in the dielectric [36]. Ceramics are typically used for DRAs due to their high permittivity and low loss. On the other hand, liquid dielectrics can be used to build a DRA. Liquid-based DRAs are a type of reconfigurable liquid dielectric antennas. They offer a number of advantages such as dynamic frequency reconfigurability by changing the liquid volume, easy access to probes, possibility to have virtually any shape, and faster prototyping.

Water is the most widely used liquid dielectric applied for liquid dielectric based DRAs. Pure water, seawater or more generally saline water with different percentages of salts are used. Even though the electrical properties of saline water are more similar to a conductor (due to a higher conductivity in the presence of salts), we will consider antennas based on this type of liquid along with antennas based on pure water.

### 2.1. Basic Design

The most common water-based antenna design is a monopole antenna shown in Figure 2. Such antenna typically consists of a cylindrical holder mounted on a dielectric substrate, a feed protruding through the substrate into the holder, and water deposited into the holder. By varying the water height in the holder, the resonance frequency of the antenna can be modified. By modifying the dimensions of the holder, the substrate, and the feed, the radiation parameters of such antennas can be optimized. Many modifications of this basic design have been proposed in the literature and will be discussed here.

It is considered that David H. Hatch was the first one to propose a so-called “ionic liquid antenna (ILA)” in 2002 [42]. The website [42] is no longer supported, but for those interested the data can be recovered using an internet archive tool [43]. The first scientific publication on the topic did not appear until 2005, when H. Fayad and P. Record presented their work at a conference [37] and later published a letter in 2006 [38]. In References [37,38], a water-based monopole antenna was proposed, where saline water was used as the dielectric material. A monopole antenna was constructed using a polyvinyl chloride (PVC) tube mounted vertically on a ground plane. A round steal electrode of diameter 0.65 mm was protruded through the base of the tube and connected to an sub-miniature A (SMA) connector. It was found that with the increase in the salinity of the water from 0 ppt to 0.8 ppt, the bandwidth of the antenna increases from 10% to 16%. However, by further increasing the salinity of the water up to 6 ppt and further, the efficiency of the antenna reduces significantly and it essentially stops resonating. The ratio of the tube diameter to the water height affects the operating frequency of such a resonator.

In Reference [39], the authors studied the performance of a DRAs based on pure water as a function of the water height. By changing the water level in a cylindrical 550×300 mm^2^ plastic holder from 134 mm to 175 mm, the resonance frequency changes from 50 MHz to 96.8 MHz. The authors confirmed that the frequency behavior agrees well with the analytical Equation (Equation 4) for the HEM_11*δ*_ (magnetic dipole) mode in a cylindrical half-space DRA [44].
(4)$$f_{0}=\frac{19.972\times 10^{8}}{2\pi a\sqrt{\epsilon_{r}+2}}\left[0.27+0.36\left(\frac{a}{2h}\right)+0.02{\left(\frac{a}{2h}\right)}^{2}\right],$$
where *h*, *a*, and $\epsilon_{r}$ are the height, radius, and permittivity of a cylindrical dielectric, respectively, and $f_{0}$ is the resonant frequency of the corresponding HEM_11*δ*_ mode. In Reference [39] the authors also found that the feed size is an important parameter, that is, the feed should be shorter than the water depth *h* and comparable to *h* for high and low h/a ratios, respectively.

Later on, L. Xing et al. further investigated the sea water monopole antenna in terms of the effects of the dielectric layer between the water and the ground and the conductivity of the water [45]. It was found that the dielectric layer has a significant impact on the resonant frequency and radiation efficiency of such an antenna, that is, smaller permittivity and higher thickness will provide better radiation efficiency. They also further studied the effect of the water conductivity (the percentage of salt). It turned out that depending on the conductivity of the water, such antenna can be regarded as a dielectric antenna, a conducting antenna or a combination of the two. In a following paper [40], the authors further investigated the radiation characteristics of this antenna with respect to the ambient temperature, diameter and height of the water layer, and the size of the feed and the ground plane. It was shown that (1) height of water and the diameter of the cylinder greatly affect the resonance frequency and the radiation bandwidth (so by properly choosing a water height to diameter ratio, the fractional bandwidth and radiation efficiency can be maximized); (2) to maximize the fractional bandwidth, the ground plane has to be at least one wavelength long; (3) the position and length of the feed does not affect the radiation characteristics of the antenna if the water conductivity is more than 4 S/m, however for less conductive water, these parameters are important [40].

For the holder of the liquid dielectric based monopole antenna, PVC or other off-the-shelf tubes are typically used which come in standard dimensions. With the help of a 3-D printer it is possible to rapidly manufacture any water holder for the antenna with optimized dimensions. In Reference [41], a low-cost monopole antenna is realized where a 3-D printer was used to fabricate the water holder.

Table 1 summarises the performance of the liquid-based monopole antennas reviewed in this subsection. This simple design offers continuous frequency reconfiguration ability for a monopole antenna, while also minimizing the antenna dimensions, with the efficiency of at least 40%.

### 2.2. Feeding Technique Variations

The basic monopole design shown in Figure 2 can be modified in a number of ways, for example, by changing the configuration of feeding. Figure 3 illustrates several feeding design variations.

In Reference [46], a so-called top-loaded probe was used to excite a saline water based monopole antenna, as illustrated in Figure 3a. This probe configuration improves the excitation of the TM mode. An optically transparent monopole antenna was constructed for maritime wireless communications using a clear acrylic ($\epsilon_{r}\;=\;2.7$) cylinder. The operating frequency can be tuned over a wide frequency range of 62.5–180.2 MHz with the bandwidth of 26.1–49.2% and radiation efficiency of 50.2–72.3%.

In Reference [47], a water-based frequency reconfigurable antenna was realized with a small cone probe loaded with a water container, similar to the one illustrated in Figure 3b. This conical probe was used to excite ${\mathrm{TE}}_{211}^{y}$ and ${\mathrm{TE}}_{121}^{x}$ modes of a water loaded rectangular holder for omnidirectional beam pattern with a wide bandwidth [48]. By varying the water height, the operating frequency was shifted from 700 MHz to 165 MHz. The efficiency of such antenna varies between 75% to 82% while the bandwidth is around 1.2–6.3%. However, the size of the antenna and the amount of water required for tuning is large.

Another variation of a cone-fed monopole antenna was presented in Reference [49] where a larger size conical feed with spiral slots etched on the cone was used to change the radiation field from an omnidirectional (no slots) to boresight type (with slots) with a wide frequency coverage from 1.6 GHz to 5 GHz. A low loss ionic liquid (trihexyltetradecylphosphonium chloride (TPC), formula: [P(nC6H13)3(nC14H29)][Cl]) was used as the dielectric in this work. TPC has a very small electric conductivity of 0.00025 S/m and remain in a liquid state at temperatures as low as −69.8 °C.

In Reference [50], the probe was modified by introducing a one-turn spiral extension which helps to achieve a wider bandwidth. An illustration of such a feed is shown in Figure 3c. This feeding structure would be hard to realise in a traditional solid DRAs. Moreover, a metasurface lid was introduced to further improve the bandwidth and circular polarization of the antenna. Choline L-alanine was used as the ionic liquid with a wide liquid range from −56 °C to 186 °C with a low electrical conductivity. Even though the authors did not demonstrate frequency tuning, it is expected that by changing the height of the liquid such tuning can be realized.

In Reference [51], a water-based monopole antenna was fed through a conducting ring with one side connected to the ground plane and the other side connected to a Γ-shape feeding arm. This design allows the incorporation of a matching network which can be tuned to obtain a good reflection coefficient. Another feed variation was introduced in Reference [52], where a circular conducting ring was coupled around the water holder and used as a feed. A vertical strip was used to connect the feed to a microstrip line. An illustration of this feed type is shown in Figure 3d.

A water-based antenna can be excited by slot antenna, such as for example, an archimedean slot antenna illustrated in Figure 3e. In Reference [53], an archimedean spiral slot was etched into the ground plane to excite a circularly polarized radiation of a square shape water holder (${\mathrm{TE}}_{111}^{x}$ and ${\mathrm{TE}}_{111}^{y}$ modes).

Table 2 summarises the performance of the liquid-based antennas with various feeding designs as reviewed in this subsection. In general, by varying the feeding technique is is possible to excite other modes of the DRA as well as to increase the tuning range. All of the designs provide continious frequency tuning.

### 2.3. Antenna Geometry Variations

Due to the nature of liquids, they can take the shape of any given container. This property is widely used in the design of various liquid dielectric reconfigurable antennas. Figure 4 illustrates a few examples of this type of antennas with various shapes.

In Reference [54], a thin and flexible tube was used as a liquid holder instead of previously proposed rigid designs. The tube can be freely shaped into a helical antenna of various height and pitch sizes as illustrated in Figure 4a. In their work [54], the authors systematically analysed the effect of various parameters (such as the length of the probe, the substrate thickness and material, liquid height and diameter) on the radiation characteristics of such an antenna where saline water was used as the liquid. A similar design was evaluated in Reference [55] and compared with a metallic helix of an equivalent size. It was found that the impedance bandwidth of the water-based helix is reduced compared to the metal helix from 107.7% to 73.1%, while the maximum radiation efficiency is comparable (92.5% for the water-based one compared to the 98.3% for the metal-based one). It was found that by increasing the number of turns of the water arm, the higher is the maximum realized gain. Moreover, by combining two water-based helical antennas of the same dimensions but opposite wound directions, it is possible to reconfigure the antenna polarization from RHCP to LHCP.

With the introduction of a parasitic water cylinder, such water-based monopole antenna can obtain a unidirectional far-field radiation pattern [56], which is essentially a Yagi-Uda antenna. By changing the water height in both cylinders, a frequency range from 1.22 GHz to 2.08 GHz can be covered. An illustration of such antenna is shown in Figure 4b. In another variation of a Yagi antenna presented in Reference [57], a central saline water monopole is surrounded by four parasitic water filled cylinders that can act as reflectors or directors depending on the amount of liquid inside. In Reference [58], a circular monopole antenna was surrounded by three sets of seven resin-based reflectors that can be filled with water to steer the beam into three distinct directions. This antenna operates in the 5.7 GHz band. Similarly, in Reference [59], a sea-water based monopole antenna was surrounded by 12 parasitic cylinders which can be individually filled with water to realize a 360° beam steering. The proposed antenna operates within the range of 334 to 488 MHz with a radiation efficiency of more than 60%, although no frequency tuning was demonstrated. In Reference [60], a multiple-input multiple-output (MIMO) antenna based on two perpendicular dipoles was coupled with four parasitic elements (Yagi-Uda) that can be loaded with water to increase their electrical length. When not loaded, the parasitic elements act as directors, and when loaded—as reflectors. By combining loaded/unloaded parasitic elements, six distinct pattern configuration can be achieved. In Reference [61], Wang et al. proposed a methodology for analyzing sea-water based monopole Yagi-Uda antenna. It is based on the assumption that the sea-water can be treated as an imperfect conductor. By applying the 3-term theory and voltage-current relation, an analytical prediction of the radiation pattern can be obtained.

In Reference [53], a rectangular container was divided into several compartments as schematically illustrated in Figure 4c. Each compartment was gradually individually filled with pure water thus providing a greater degree of frequency tunability from 2.3 to 2.92 GHz. In Reference [62], another compartment geometry was proposed where four quadrants where used as the dielectric load. A schematic illustration of such design is shown in Figure 4d. The water level in each quadrant can be controlled individually, which affects the radiation pattern created by an L-shape probe (LHCP, RHCP, or LP).

In Reference [64], two containers were placed near a single probe. When one of them was filled with ethyl acetate ($\epsilon_{r}\;=\;6.6$), the resulting polarization was RHCP. When the other container was filled instead, the polarization changed to LHCP. The antenna operates in the range 2.31–2.72 GHz with the radiation efficiency of >70%. In Reference [65], another compartment geometry was proposed for switching between the RHCP and the LHCP. Measurement results showed that this antennas can achieve a CP reconfigurability in dual band 1.55–1.72 GHz and 2.29–2.52 GHz with wide bandwidths of 10.4% and 9.7%, respectively. In Reference [66], another compartment geometry was proposed to for a wide-band frequency reconfigurable antenna based on pure water. A bow-tie feeding slot is loaded with a central square acrylic container which is surrounded by another eight square acrylic containers. By changing the height of the liquid in these containers, different frequency states can be achieved in the range of 168–474 MHz. A high efficiency of more than 80% was achieved. The authors also analysed the effect of the slot size, ground plane size, and temperature on the radiation characteristics of the antenna.

In Reference [63], an optically transparent patch antennas was realized with pure water used for both the patch and the ground. The antenna was excited by a small disk loaded probe positioned in the air between the patch and the ground as illustrated in Figure 4e. It was demonstrated that the usable frequency band can be shifted by changing the height of the ground. In a follow up work [67], the authors modified the previous design by adding an annular water ring around the disk-loaded probe. This addition helped to reduce the size of the antenna (lower operating bandwidth) by effectively introducing an additional shunt capacitance between the patch and the ground.

In Reference [68], a low-profile T-shape water based antenna was fed by a small rectangular probe. Two parasitic water-based elements were added on both sides of the fees to improve the impedance matching. A 34% impedance bandwidth was achieved from 3.18 GHz to 4.47 GHz with the radiation efficiency up to 80%. In Reference [69], a water patch was excited by an L-shape probe. Even though the authors did not show the tunability of the system, from the model it is expected that the resonance frequency of the antenna will shift down with the gradual introduction of water. From the perspective of the paper, the main advantage of such an antenna is its transparency which allows its seamless integration with solar cells.

In Reference [70], a dipole was loaded with a water-based bow-tie shaped structure to lower its electrical length. By applying water-based bow-tie structures of different dimensions, frequency tuning within 0.9 to 1.4 GHz can be achieved. The authors also compared the difference between the pure (Im(ϵ)= 2.4–6.2) and sea water (Im(ϵ)= 40–105). From the simulations, the radiation efficiency of the pure water-based antenna is stable throughout the whole frequency range and above 95%, while for the sea water-based one the efficiency changes from 75% to 90%. In this work, the water-filled bow-tie structures of different dimensions were fabricated separately and then installed one after another. It is a very slow reconfiguration process, and a more practical one could be to use a flexible PDMS-based water holder which can be gradually stretched.

Table 3 summarises the performance of the liquid-based antennas with various geometry designs. It can be seen that liquid dielectrics can be used to realise many types of antenna geometry that provide discrete or continuous frequency tuning, radiation pattern and polarization reconfiguration.

### 2.4. Other Liquids

Even though water is so widely used as a dielectric liquid of choice for frequency tuning, it has a number of limitations such as low radiation efficiency at higher frequency bands (>1 GHz) due to a frequency dependant permittivity [71] and small working temperature range (due to the phase change below 0 °C and above 100 °C). These drawbacks limit the application of water-based antennas. However, other dielectric organic solvents can offer smaller loss tangent and lower freezing point.

In particular, in Reference [49], an alternative ionic liquid was used as a dielectric loading for a linearly polarized monopole antenna. The liquid of choice was trihexyltetradecylphosphonium chloride (TPC) (formula: [$P{\left(nC_{6}H_{13}\right)}_{3}\left(nC_{14}H_{29}\right)]\left[Cl\right]$) which retains its liquid state at temperatures as low as −69.8 °C and has a small loss tangent of <0.001. Such ionic liquids are meant to replace water as the commonly used dielectric liquid of choice. In the proposed design the ionic liquid lowered the resonance frequency from around 7 GHz to around 4 GHz and improved the fractional bandwidth (from 20.7% to 44%) of a monopole antenna. In Reference [50], another ionic liquid was used called Choline L-alanine. It also has a wide liquid range from −56°C to 186°C with a low electrical conductivity (σ=0.00021 S/m). This ionic liquid was used for a liquid-based monopole antenna. Even though the authors did not demonstrate frequency tuning, it is expected that by changing the height of the liquid such tuning can be realized. Electrical properties of other ionic liquids that can be used for antenna design were systematically studied in References [72,73]. A recent book [74] on modern functional organic liquids my be helpful for further improvement of the liquid reconfigurable antennas.

### 2.5. Liquid Solutions and Other Combinations

A combination of a liquid dielectric with a high permittivity solid dielectric powder can help to control the effective permittivity and to form a liquid dielectric with desired characteristics. For example, in Reference [75], colloidal barium strontium titanate (BSTO formula: $Ba_{0.6}Sr_{0.4}TiO_{3}$, ϵ∼500, tanδ∼0.01) was dispersed in hydrotreated naphthenic oil (ϵ=2.1, tanδ=0.001). This colloidal dispersion was used to provide a frequency tuning for liquid-based monopole antenna with a tuning range between 2.75 GHz to 4.22 GHz. A practical range of the liquid height between 8 mm and 24 mm was found that provides good 2:1 SWR ratio. Outside these limits, the monopole parameters must be changed to maintain an acceptable performance, for example by having a matching network [39] or variable height probe.

A combination of two or more liquids can be used to improve the radiation characteristics even further. For example, in Reference [76], a combination of pure and saline water was used to improve the bandwidth of a monopole antenna. In the frequency range from 50 MHz to 300 MHz, sea water can be considered as a good conductor. On the other hand, the pure water can be considered as an imperfect dielectric [46]. As was suggested in Reference [77], a dielectric ring around a monopole antenna can improve its bandwidth. In Reference [78], it was demonstrated that by placing a dielectric ring of pure water around a seawater monopole antenna, the bandwidth can be improved by 73% compared to a single sea water antenna. By changing the shape of the dielectric ring, the bandwidth can be improved even further, resulting in a frequency range from 54.5 MHz to 251.4 MHz (129%).

For some applications the high freezing temperature of water can become an issue. That is why some researches are exploring other alternatives such as sea water (mixture of water with salts, for example, NaCl or KCL) and mixing water with liquids that have lower freezing point, such as ethanol. However, ionised water has a significant conductivity, for example, sea water has the conductivity of around 4.7 S/m. Thus, one of the main concerns of using ionised water for liquid frequency tuning is the efficiency [41].

Another solution is to mix water with propylene glycol (PG) which is a kind of commonly used antifreeze that has good solubility in water. The freezing point of water with 5% PG is ∼3 °C which meets the lower limit for temperature range of commercial electronics (0–85 °C). In Reference [79], the complex permittivity of a liquid solution of water and PG was investigated numerically and experimentally as a function of frequency (0–18 GHz), concentration (0∼70%) and temperature (−10∼70 °C).

In Reference [80], a DRA is realized with a combination of solid and liquid dielectric materials. The basic monopole DRA is made of a K9 glass (ϵ=6.85) and the puter zone can be filled with a liquid dielectric ethyl acetate (ϵ=7.1). The inner glass-based DRA is excited in its broadside ${\mathrm{HEM}}_{11\delta }$ mode, and it is switched to the conical ${\mathrm{TM}}_{01\delta }$ when ethyl acetate is added. This antenna operates in the range of 3.75–5.37 GHz with a 35.5% impedance bandwidth, and with a >80% efficiency in the first state and 50–63% efficiency in the second state.

Table 4 summarises the liquid-based antennas that utilise other liquid types and liquid solutions. By carefully choosing the liquid type, antennas performance can be improved, in particular in terms of the lower operation temperature and losses.

## 3. Liquid-Assisted Antennas

In the previous Section 2, liquid was the essential element of an antenna, acting as a radiator or even reflector. Moreover, all of the discussed antenna designs were three dimensional and often required significant amount of liquid to change the radiation behavior. In this section, other types of antenna design will be considered that have a dielectric liquid in the immediate vicinity of metal radiating structures. In these designs, small amounts of liquid modify local currents of an antenna to alter its radiation performance. In most of the designs, microfluidic channels are introduced in the substrate between the ground and the top layers. By pumping a liquid dielectric through strategically placed channels, a controlled frequency tuning can be achieved.

### 3.1. Microfluidic Channels

A planar monopole antenna is a variation of a monopole antenna that is printed on a dielectric substrate. It consists of a quarter wavelength $\lambda_{g}$ strip line and a ground plane printed on the top and bottom of the substrate, respectively, where $\lambda_{g}$ is the guided wavelength in microstrip environment. Such antenna produces an omnidirectional radiation pattern. The guided wavelength $\lambda_{g}$ depends on the permittivity of the substrate. In Reference [81], the permittivity of the substrate was modified locally by introducing channels that are perpendicular to the monopole stripline. The channels are integrated into the substrate closer to the monopole end where the electric field is strongest. By gradually filling four channels with water, a number of discrete frequency states can be achieved. A schematic of such frequency reconfigurable antennas is shown in Figure 5a. When all the channels are vacant, the center frequency is around 5.5 GHz. By gradually filling the channels, starting from the one on the far end, the center frequency shifts to 4.98 GHz, to 4.76 GHz, and finally to 4.48 GHz. The corresponding impedance bandwidth is gradually changing from 36.9% to 20.7%.

Patch antenna is a widely used type of a planar antenna that consists of a square (rectangular) patch and a ground plane placed on top and bottom of a dielectric substrate, respectively. The patch can be fed through a via at the center or through a microstrip line. The fringing electric fields at the edges of the patch are responsible for the radiation of the antenna. Moreover, the radiation characteristics can be modified by changing the height and permittivity of the dielectric substrate. If these two properties of the substrate can be modified in real time, instant frequency tuning can be achieved.

In Reference [82], several frequency states are introduced to a square patch antenna with the help of four microfluidic channels introduced between the patch and the ground. The channels were strategically placed near the radiating edges of the patch, where the electric field amplitude is maximum in the ${\mathrm{TM}}_{100}^{z}$ mode, as illustrated in Figure 5b. Each channel has two states: (1) air and (2) water. By mixing these two sates between the four channels, several frequency states can be achieved. The switch between frequency states is not continuous but discrete that is why the authors called it “frequency hopping”. The achieved frequency hopping range is from 1.4 GHz to 1.9 GHz. In general, if the number of channels is *N*, then the number of different frequency states is $S\;=\;2^{N/2}$ [82]. Moreover, Tang et al. showed that the same principle can be applied to an array of patch antennas, for example, a 4×1 patch antenna array.

In Reference [83], a similar idea was realized on a patch antenna with two microfluidic channels placed near the patch edges. By changing the type of liquid (among water, acetone, methanol, and ethanol) inside the channels, frequency tuning was realized from 16.8 GHz to 28.4 GHz.

In Reference [84], the performance of a patch antenna was altered with the help of four small cylindrical containers placed like pillars between the ground and the patch, as shown schematically in Figure 5c. Because the electrical field between the patch and the ground plane is stronger than the one above the patch, it is more efficient to put the pillars there, as the authors demonstrated. The containers were strategically placed at the four corners of the patch to be able to control the type of polarization produced by patch. By changing the water level inside the cylinders, frequency and polarization tuning can be achieved.

In Reference [85], a slot antenna was used with two channels filled with air, water or acetone to independently control the first and second resonant modes. The location of the E-field minimum and maximum of a slot antenna resonator depends on its mode. In particular, the location of the E-field maximum of the second mode is the same as the E-field minimum of the first mode. It means that by strategically placing the channels, it is possible to separately tune the two modes, as illustrated in Figure 5d. The radiation efficiency is relatively low, however, if acetone is used instead of water, the radiation efficiency can be improved [85].

In Reference [87], a liquid dielectric FC-40 ($\epsilon_{r}$ = 1.9, tanδ = 0.0005) was used to move a metallized plate inside a microfluidic channel, thus changing the resonant frequency of a planar monopole antenna. A continuous tuning form 1.7 to 3.5 GHz can be achieved with the tuning speed of 1565 MHz/s. The power handling of such antenna was improved compared to a previous work [88], where a liquid metal was used instead of the metallized plate. Specifically, it was found that this antenna can operate with 15 W of continuous power. The further power increase is limited by the maximum temperature handling of the micro-pumps used to move dielectric liquid.

In Reference [89], the resonance frequency of a patch antenna was modified by pushing two metallic rods inside a microfluidic channel with the help of a mineral oil. Two microfluidic channels were made in a photopolymer resin ($\epsilon_{r}\;=\;3.2$, tanδ=0.05) that was 3D printed. Two bras cylinders were inserted into the channels and could be moved along the non-radiating patch edges by injecting oil through a syringe. Frequency tuning in two closely spaced bands was demonstrated.

In Reference [90], a liquid-assisted switch was proposed for a Vivaldi antenna that function in two operating bands: from 2.5 GHz to 3.25 GHz and from 3.25 GHz to 4.50 GHz. The switch was introduced at the back slot such that when the liquid is introduced in the switch, it shortens the current path and thus results in a higher operating frequency. The liquid of choice was a 2 mol KCl solution that was found to provide the highest efficiency of up to 87%.

In Reference [86], a microfluidic water-based switch was implemented for a polarization reconfigurable patch antenna. The patch consists of two orthogonally overlapped strips that are electrically isolated at their overlap by four gaps, as illustrated in Figure 5e. A microfluidic channel is implemented around the overlap. By capacitively loading the gaps with water and air in an alternating manner, two antenna states can be activated independently. With a center frequency around 2.4 GHz, fractional bandwidth of 2.19–2.39%, and efficiency of 22% this antenna can operate in two perpendicular polarization states.

Table 5 summarises liquid-assisted antennas that utilize microfluidic channels to control the current distribution. Unlike the liquid-based antennas, liquid-assisted antennas with microfluidic channels mostly provide discrete tuning with only several states.

### 3.2. Cavities

Another approach to changing the effective permittivity of the substrate is to introduce a cavity into it, which can be filled with a dielectric material. The substrate then can be rigid or flexible. For example, in Reference [91], a rectangular cavity between a patch antenna and the ground was filled with a low-loss transformer oil ($\epsilon_{r}=2.24$, tanδ = 0.005). By varying the height of the oil in the cavity, the volume ratio of air to liquid (oil) changes, as illustrated in Figure 6a, thus changing the effective permittivity of the substrate. The achieved tuning range is from 1.42 GHz to 1.96 GHz with a high radiation efficiency due to the low-loss transformer oil.

In Reference [92], a planar spiral antennas made of liquid metal incorporated into a flexible PDMS membrane was proposed. The membrane is positioned over a rigid planar ground plane and fixed with an annular FR-4 substrate. A microstrip feed is coupled to both arms of the spiral. When the air pressure under the membrane changes, it inflates the structure to achieve a spherical cap, as illustrated in Figure 6b. A tuning range of 526–542 MHz was achieved with this technique. However, the efficiency of such antenna is very low at the planar state, with the maximum efficiency of 55% reached at the fully inflated state.

In Reference [93], a millimeter-wave microstrip patch antenna was printed on an ultrasoft elasotometric polydimethylsiloxane (PDMS) substrate with a membrane, such as in the illustration on Figure 6c. The technological feasibility of PDMS membranes as millimeter-wave antenna substrates was previously demonstrated in Reference [97]. The main technological difficulty of fabricating metal patterns on PDMS is its high thermal expansion coefficient and weak adhesion to metal [98,99]. Thus, PDMS cannot be used in conventional evaporation and sputtering metallization techniques. In Reference [97], the authors suggested a transfer technique with the help of a Ti + SiO_2_ adhesion layer that mitigates the aforesaid issues. With the help of pneumatic actuation, air can be introduced into the cavity and thus change the distance between the patch and the ground. By changing the distance between the ground and the patch, a continuous change in the resonant frequency can be obtained, that is, from 55.35 GHz to 51 GHz (8%) with the dimensions as in Reference [93]. In this experiment, the cavity was filled with air using an electronic syringe pump such that the total maximum air volume was 55μL. The matching level of less than −17 dB and stable radiation characteristics were achieved at all resonance positions [93]. A 4×2-element microstrip antenna array based on this technology was also demonstrated [100]. However, the authors did not provide any data on the stability of the electrical conductivity of the patch antenna after several cycles of inflating/deflating.

In Reference [94], a similar antenna design was suggested. In this work, multi-wall carbon nanotubes (MWNT) were used to facilitate the adhesion of gold to the PDMS substrate. A 16 × 16 mm^2^ square patch antenna was simulated and fabricated. When air was injected into the cavity, the distance between the patch and the ground changed, resulting in the frequency shift from 6.1 GHz to 5.7 GHz. Moreover, the authors conducted stretchability tests where the resistance of the patch was measured at each height change as the air was injected into the cavity to verify the repeatability of the experiment.

Similar to Reference [93], the resonance of a patch antenna can be changed by reshaping the the volume of the dielectric between the patch and the ground [95]. In Reference [95], K. Noda et al. came up with a liquid-encapsulation technique that allows to deposit polymer directly on a liquid surface. The encapsulated liquid can then be deformed using electrostatic force. This method allows to use liquids as a mechanically tunable dielectric layer, instead of air. As liquids have higher permittivity, the antenna size can be efficiently reduced [95].

In Reference [96], a similar idea is realised on a polyurethane (PU) substrate. Compared to PDMS, PUs can be more easily metalized due to a lower thermal expansion coefficient. Moreover, the Young’s modulus of PUs is higher, thus PUs are more stiff compared to PDMS which means better mechanical stability but less stretchability [96]. The fabricated patch antenna had the dimensions of 30×30 mm^2^ and demonstrated a 32% efficiency and a 3.9% frequency agility. In Reference [101], it was proposed that by patterning the metallic patch (e.g., as in Reference [102]), the flexibility of such antenna can be improved.

In Reference [103], the resonance frequency of a bent dipole antenna was modified by the presence of castor oil ($\epsilon_{r}\;=\;2.7$) or ethyl acetate ($\epsilon_{r}\;=\;6$). The antennas was incorporated into a sealed plastic box (acrylonitrile butadiene styrene (ABS) body and clear polycarbonate lid $\epsilon_{r}∼2.7$). A flexible diaphragm made of a 1 mm latex rubber sheet was placed across the box. One of the liquids was pumped in bounded by the diaphragm which expanded as the volume of liquid was gradually increased such that the diaphragm touches the whole enclosure eventually. Due to the presence of the latex diaphragm the antenna parts and the ABS box are protected from the direct contact with the liquid. A frequency tuning range of 0.9 to 1.44 GHz was demonstrated with ethyl acetate with a high radiation efficiency of 69%–92%.

Table 6 summarises the liquid-assisted antenna designs that utilize cavities inside the substrate for radiation control. As in this case the liquid can be gradually introduced into the cavity, a continuous tuning can be achieved.

### 3.3. Other Tuning Techniques

Other design variations proposed for antenna reconfiguration that are based on liquid dielectric materials but did not fit in the previous sections are discussed here.

#### 3.3.1. Static vs. Dynamic

One of the advantages of the liquid dielectric antennas is their rapid reconfigurability as the liquid can be quickly removed when the antenna is not is use which lower the weight of the antenna and ease its transportation. It also means that a dynamic type antennas can be realized. For example, in Reference [51], a shunt excited monopole antenna was positioned vertically on the sea surface and the bottom end of the cylinder was connected to a water pump. When the pump was on, it would take the sea water, guide it through the cylinder, and thus activate the antenna. It was found that the water stream coming out of the top of the cylinder increases the bandwidth of the antenna. However, such design proved only two states (on and off) and no frequency tuning.

A variation of this dynamic type antenna was presented in Reference [104], where this monopole antenna was tilted with respect to the vertical axis such that the sea water stream coming out of the top of the cylinder forms a sinusoidal curve. Again, no frequency tuning was demonstrated, however, it is possible that by changing the tilt angle the resonance frequency may shift. In Reference [105], a similar tilted design was demonstrated to receive a broadcast signal.

#### 3.3.2. Gravitational Tuning

Due to the nature of liquid, it can conform to any shape of the vessel in such a way that its potential energy is minimized. It means that under normal conditions the surface of liquid is flat and perpendicular to the direction of gravity. This gives an opportunity for another degree of tunability. For example, in Reference [26], the authors proposed to use gravity for frequency tuning of metasurfaces. In one of the examples, a partially filled elliptical cylindrical container was used as a meta-cell. When its orientation changes, the liquid inside is redistributed due to gravity such that the effective shape of the metasurface changes which results in a different transmission response. This gravitational technique can be applied for the liquid-based antenna design as well. For example, in Reference [106], a cylindrical DRA filled with two types of liquids was used to stir the radiation pattern. The advantage of such technique is that no liquid pumping is required unlike in any of the previously discussed designs.

#### 3.3.3. Thermal Tuning

The dielectric permittivity of water depends on the ambient temperature and this allows to use temperate as another mean for frequency tuning [71]
(5)$$\epsilon \left(\nu ,T\right)=\epsilon_{\infty }\left(T\right)+\frac{\epsilon_{0}\left(T\right)-\epsilon_{\infty }\left(T\right)}{1-i2\pi \nu \tau (T)},$$
where *T* is the water temperature (°C), ν is the frequency of the electromagnetic field (Hz), τ is the characteristic time of the relaxing/resonating system (s), and $\epsilon_{\infty }$ is the water permittivity at an “infinite” frequency [71]. For example, in Reference [107], the frequency response of a dipole antenna was manipulated by changing the local temperature of a water-based substrate.

## 4. Discussion

The main advantage of liquid dielectric materials is their fluidity and conformability. These two properties are widely used in the design of reconfigurable antennas. In the case of liquid-based antennas, where a dielectric liquid is used as the main radiating structure, the antenna geometry can take any shape due to its conformability. Complex shapes are hard to realize with conventional solid DRAs, so liquid dielectric antennas can offer a competing advantage and also provide reconfigurability. Liquid dielectrics offer an easy reconfiguration technique, where by changing the height, volume, or shape of the liquid, radiation characteristics of an antenna can be modified. High permittivity dielectric liquids, such as water ($\epsilon_{r}∼80$), can help to reduce the size of an antenna. However, water is lossy at higher frequencies (>1 GHz) which reduces the radiation efficiency. Adding salts (NaCl, KCl) to water will increase the conductivity and widen the bandwidth. The permittivity of water depends on the ambient temperature, which can be used as another tuning mechanism or can become a limitation (narrow temperature range of operation). Other organic liquids were proposed that have a wider temperature range of operation. Oils are less lossy and thus can have a better radiation efficiency, however, their permittivity is much smaller than that of water, which will affect the tuning range. Many liquid dielectric materials discussed in this review are easily available and low cost. Moreover, most of these liquids are optically transparent which means that liquid antennas based on them can be seen as optically transparent as well and could be integrated with other components, such as solar cells, on a single platform.

In the case of the liquid-assisted antennas, liquid dielectric materials are used to modify the local currents to reconfigure a metal-based antenna. The most crucial thing for a liquid-assisted reconfigurable antenna is the placement of the liquid holders, for example, microfluidic channels or cavities. The surface current and electrical field distribution of an antenna under consideration have to be modeled and analysed. Then the channels/cavities need to be placed near the location(s) of the current/electric field maximum to maximise the tuning range. It is even possible to independently control two resonant modes by strategically placing the channels. Radiation pattern can be reconfigured by placing the channels near the radiating edges of an antenna.The higher the permittivity of the dielectric liquid, the wider frequency tuning range can be achieved. If microfluidic channels are used, only discrete reconfiguration states can be achieved. On the other hand, if a cavity is used, it is possible to have a continuous reconfiguration (tuning). All liquid-assisted antennas reported in the literature up to date are planar unlike the 3-D liquid-based ones. This is because liquid dielectric is used only as a reconfiguration mechanism while there is a main radiating part. This also means that the amount of liquid dielectric material required for liquid-assisted antennas is smaller compared to the liquid-based ones.

Based on the analysis of the papers considered in this review, the designs with frequency, pattern, and polarization reconfiguration constitute 74%, 18%, and 8%, respectively, of the total number of papers under review. Most of the reconfiguration designs are for frequency reconfiguration. It demonstrates how liquid dielectric materials are particularly useful for frequency reconfiguration techniques due to their fluidity. The major part of the discussed designs fall into the category of liquid-based antennas (65% of the total number of papers under review). This demonstrates how the conformability and fluidity of liquid dielectric materials help to realize many different antenna shapes and designs.

The topic of liquid dielectric reconfigurable antennas is relatively new and some of the practical and implementation issues have not been resolved yet. One of the main limitations of this kind of reconfigurable antennas is their low switching speed. Indeed, compared to other commonly used techniques, such as RF-MEMs and PIN diodes with their switching speeds of about 1–200μsec and 1–100 nsec, respectively, the switching speed of most of the discussed liquid-based techniques is on the order of magnitude of one second. Moreover, switching speed of liquid dielectric reconfigurable antennas is rarely discussed in the papers, even though this is a very important parameter when evaluating antenna reconfiguration. For a liquid dielectric reconfiguable antenna, switching speed primarily depends on the method used to obtain different liquid dielectric volume (height, type) and in most of the works it is done manually with a measuring tool like a syringe (when the method is not mentioned it is assumed to be manual). Some authors used micro-pumps [46,47,51,52,80,82,87,90,93,94] and only one group demonstrated a closed loop system with a microcontroller and a mirco-pump [52]. Only these few papers mentioned the switching speed of their proposed reconfigurable antennas [80,87,92].

Other important parameters for reconfigurable antennas are the reliability and repeatability of the results. It is a measure of the accuracy and consistency of the measurements when the amount of liquid is repeatedly changed inside a container, microfluidic channel or a cavity. A good reliable antenna needs to be able to handle many cycles of liquid dielectric refills and still give consistent predicted results within a certain minimal error. However, only two papers mentioned this important parameter when evaluating their proposed designs [64,87,92,94].

DC power consumption is another important practical issue. All the additional mechanisms (e.g., a micro-pump), that are used to reconfigure an antenna, consume additional power besides the amount of power consumed by the antennas itself. A high power consumption will limit the maximum operation time of the antenna if the power supply is a portable battery. However, DC power consumption has only been stated in one work considered in this review [80]. It should be noted that the DC power needed for PIN-diode-based switches has to be kept on during the whole time. On the other hand, the pump switching system has to be kept on only when the antenna needs to be switched to another configuration. These differences need to be taken into consideration when estimating the total power consumption of such antennas, which can be especially important for mobile applications.

Another important parameter related to power is the RF power-handling capability. One of the advantages of liquid reconfigurable antennas is their highly linear behaviour which means they are expected to remain linear under high-peak-power excitations [9]. However, power-handling capability of liquid dielectric reconfigurable antennas was only discussed in one paper [87].

The total size and weight of a resulting liquid dielectric reconfigurable antenna can be another practical issue. Once all the necessary additional components (such as micro-pumps, tubing, etc.) are integrated, the total size and weight of the antenna will increase. However, the implementation of the micro-pumps and tuning for liquid dielectric material delivery is easier than implementing a DC biasing network for a PIN diode-based switches.

Generally, it is not easy to compare conventional (RF-MEMs and PIN diodes) and liquid dielectric approaches and there is very little work published directly comparing the two. In Reference [86], the authors compared two polarization reconfigurable patch antennas of similar designs. In one of them, the reconfiguration mechanism was realized with microfluidics and in the other one—with PIN diodes. These two antennas were compared side by side in terms of electrical size, radiation efficiency, radiation pattern, and switching speed. It was found that even though the switching speed of the liquid reconfigurable antennas is much slower (∼sec) than that of the conventional PIN diode-based ones (∼μsec), liquid reconfiguration approach offers advantages in terms of the higher radiation efficiency (22% compared to 10% in the PIN diode case), longer electrical length, and zero DC interference [86]. It means that liquid dielectric reconfigurable antennas can be useful in application where the switching speed is of less importance than the antenna performance.

## 5. Conclusions

This paper summarized state-of-the-art techniques and methods used to design, optimize, and apply liquid dielectric reconfigurable antennas. Communication systems that rely solely on stationary antennas or a combination of those are no longer suitable for the evolving needs of the modern communication technology where the physical size of the hardware becomes increasingly small. Liquid dielectric materials offer a novel mechanism for radiation control and reconfigurability, worthily complementing the existing reconfiguration methods. Two main approaches were identified for the antenna reconfiguration using liquid dielectrics, namely liquid-based and liquid-assistive techniques. In the liquid-based technique, a liquid dielectric material in itself is used as the main radiating structure. In the liquid-assisted technique, a liquid dielectric material is used to modify local currents of an existing metal-based antenna to reconfigure its radiation parameters. This paper discussed the main design principals, advantages and limitation of this type of reconfigurable antennas. Even though there are a few technological and practical challenges that have to be solved, liquid dielectric reconfigurable antennas offer higher efficiency and wider operational bandwidth compared to traditional reconfiguration mechanisms based on electrical switches.

## Figures and Tables

**Figure 1 materials-13-01863-f001:**
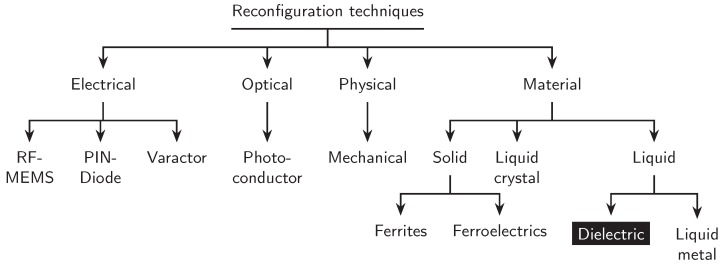
Categorization of antenna reconfiguration techniques [4].

**Figure 2 materials-13-01863-f002:**
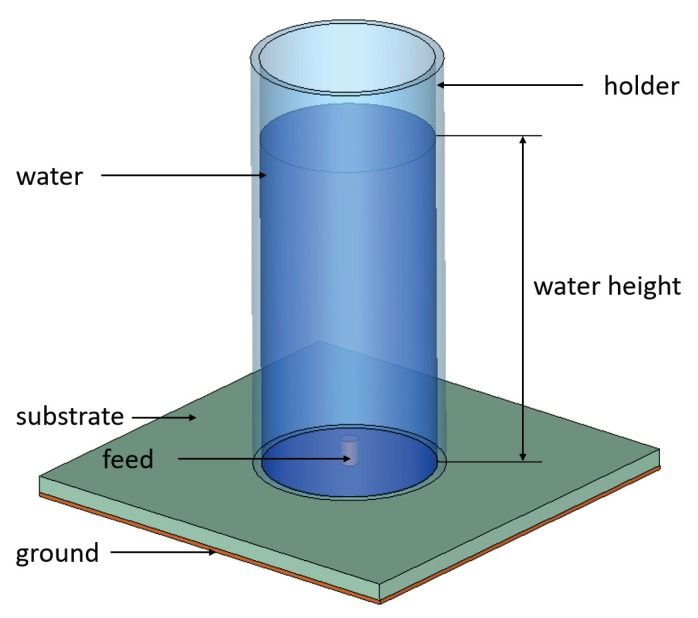
Schematic of a basic water-based monopole antenna [37,38,39,40,41].

**Figure 3 materials-13-01863-f003:**
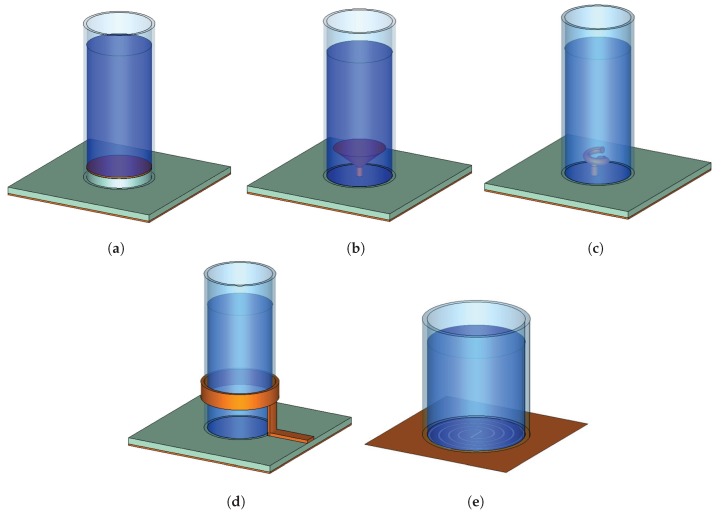
Monopole antenna with different feeding probes. (**a**) [46], (**b**) [47,48,49], (**c**) [50], (**d**) [51,52], (**e**) [53]].

**Figure 4 materials-13-01863-f004:**
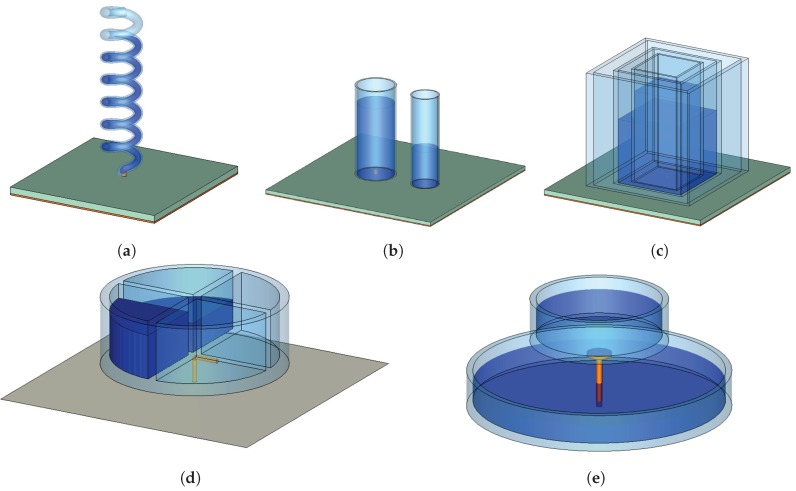
Liquid dielectric based antennas of various geometries. (**a**) [54,55], (**b**) [56,57,58,59,60,61], (**c**) [53], (**d**) [62], (**e**) [63]].

**Figure 5 materials-13-01863-f005:**
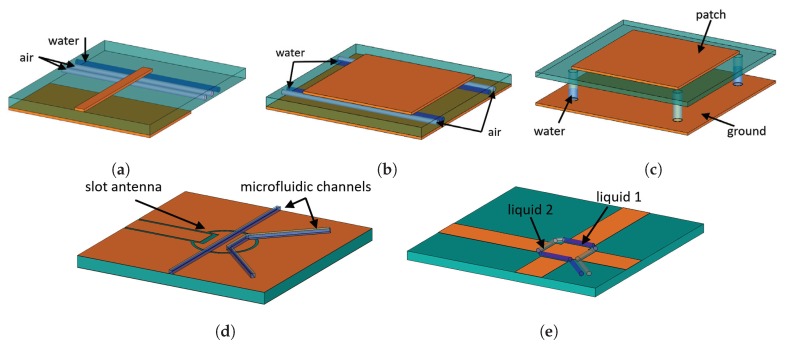
Planar antennas with microfluidic channels. (**a**) [81], (**b**) [82,83], (**c**) [84], (**d**) [85], (**e**) [86].

**Figure 6 materials-13-01863-f006:**
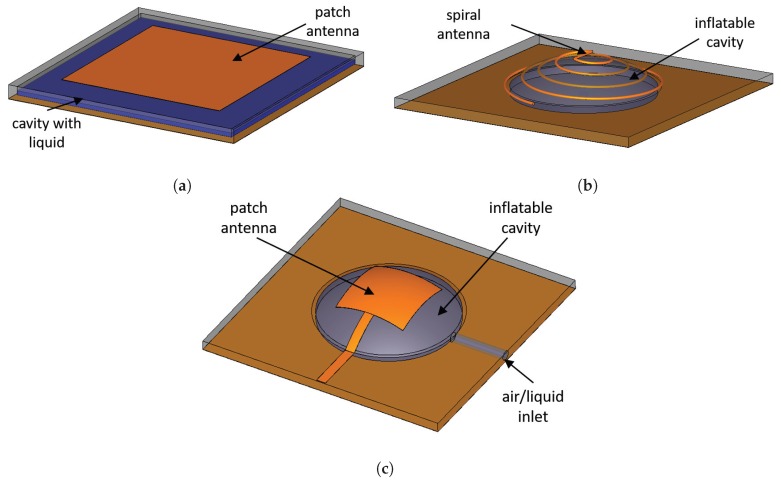
Planar antennas with a cavity. (**a**) [91], (**b**) [92], (**c**) [93,94,95,96].

**Table 1 materials-13-01863-t001:** Basic liquid-based monopole antennas.

Ref.	Antenna Type	Pla-Nar	Dimensions mm^3^$\left(\lambda_{0}\right)$	Reconfig. Type	Liquid Type	Tuning Type	Frequency Range (GHz)	TR Equation (Equation 1)	TX Equation (Equation 3)	Radiation Efficiency
[38]	mono- pole	no	21×25×25$(0.13\lambda_{0}$)	frequency	saline water	contin.	1.597–1.604	-	-	40%
[39]	mono- pole	no	550×550×300$(0.18\lambda_{0}$)	frequency	pure water	contin.	0.050–0.097	64%	1.94	N/A
[40]	mono- pole	no	25×25×100 $\left(0.63\lambda_{0}\right)$	frequency	saline water	contin.	1.40–1.88	29%	1.34	46–58%
[41]	mono- pole	no	35.4×11×11 $\left(0.54\lambda_{0}\right)$	frequency	saline water	contin. *^a^*	2.29–4.58	67%	2	40–70%

*^a^* not explicitly demonstrated.

**Table 2 materials-13-01863-t002:** Feeding Technique Variations.

Ref.	Feed Type	Pla-Nar	Dimensions mm^3^$\left(\lambda_{0}\right)$	Reconfig. Type	Liquid Type	Tuning Type	Frequency Range (GHz)	TR Equation (Equation 1)	TX Equation (Equation 3)	Radiation Efficiency
[46]	disk	no	1050×100×100 $\left(0.63\lambda_{0}\right)$	frequency	sea water	contin.	0.063–0.180	96%	2.86	50.2–72.3%
[47]	cone	no	300×300×200 $\left(0.70\lambda_{0}\right)$	frequency	pure water	contin.	0.165–0.700	124%	4.24	75–82%
[49]	cone	no	100×100×41 $\left(2\lambda_{0}\right)$	frequency	ionic liquid	contin. *^a^*	1.6–5.0	103%	3.125	N/A
[50]	spiral *^b^*	no	100×100×41 $\left(0.93\lambda_{0}\right)$	frequency	ionic liquid	contin. *^a^*	a few bands between 1.37 and 2.8	-	-	N/A
[51]	ring	no	1000×330×110 $\left(0.33\lambda_{0}\right)$	frequency	sea water	contin.	0.065–0.099	42%	1.52	47.0–78.1%
[52]	ring	no	30×30× $\left(0.46\lambda_{0}\right)$	frequency	saline water	contin.	3.5–4.6	27%	1.31	10–80%
[53]	slot	no	350×350×150 $\left(0.47\lambda_{0}\right)$	frequency	pure water	contin.	0.155–0.400	88%	2.58	90%

*^a^* not explicitly demonstrated in the paper, *^b^* without the metasurface lid.

**Table 3 materials-13-01863-t003:** Antenna Geometry Variations.

Ref.	Antenna Type	Pla-Nar	Dimensions mm^3^ $\left(\lambda_{0}\right)$	Reconfig. Type	Liquid Type	Tuning Type	Frequency Range (GHz)	TR Equation (Equation 1)	TX Equation (Equation 3)	Radiation Efficiency
[54]	helical *^a^*	no	70×100×185 $\left(1.5\lambda_{0}\right)$	freq., rad. pattern	saline water	contin. *^b^*	1.70–2.44	36%	1.44	N/A
[55]	helical	no	180×70×70 (1.77λ)	polariza- tion	pure water	discrete (N = 2)	1.17–2.95	86%	2.52	40–80%
[63]	patch	no	299×299×37.6 $\left(3\lambda_{0}\right)$	frequency	pure water	contin.	1.8–3	50%	1.67	≤82%
[67]	patch	no	154×154×30.5 $\left(1.2\lambda_{0}\right)$	frequency	pure water	contin. *^b^*	1.5–2.4	46%	1.60	≤78%
[56]	Yagi-Uda	no	58×32×10 $\left(0.39\lambda_{0}\right)$	freq., rad. pattern	water	contin.	1.22–2.08	52%	1.71	75%
[57]	Yagi-Uda *^c^*	no	350×350×200 $\left(1.69\lambda_{0}\right)$	freq., rad. pattern	saline water	contin.	0.64–1.45	78%	2.27	>60%
[58]	circular patch	no	60×60×20 $\left(1.15\lambda_{0}\right)$	rad. pattern	pure water	discrete (N = 3)	5.68–5.72	-	-	N/A
[61]	Yagi-Uda	no	102×90×30 $\left(0.53\lambda_{0}\right)$	rad. pattern	sea water	discrete (N = 2)	1.08–1.55 1.00–1.46	36% 37%	1.44 1.46	50% 51%
[60]	MIMO	yes	200×200×7.8 $\left(0.63\lambda_{0}\right)$	rad. pattern	pure water	discrete (N = 6)	0.85–0.95	-	-	>60%
[62]	loaded probe *^d^*	no	50×50×37 (0.49λ)	polariza- tion	pure water	contin. *^b^*	2.30–2.92	23%	1.27	58–74%
[65]	cross- shaped	no	47×47×49.1 $\left(0.41\lambda_{0}\right)$	polariza- tion	ethyl acetate	discrete (N = 2)	1.55–1.71 2.29–2.52	10% 9.6%	1.10 1.10	74%
[66]	loaded bow-tie slot	no	278×278×131 $\left(0.44\lambda_{0}\right)$	frequency	pure water	contin.	0.168 - 0.474	95%	2.8	>80%
[53]	loaded spiral slot	no	350×350×150 $\left(0.47\lambda_{0}\right)$	frequency	pure water	contin.	0.155–0.400	88%	2.6	up to 90%
[64]	loaded probe	no	120×120×34 $\left(1.09\lambda_{0}\right)$	polariza- tion	ethyl acetate	discrete (N = 2)	2.31–2.72	-	-	>70%
[68]	T-shape loaded probe	yes	60×29×4 $\left(0.89\lambda_{0}\right)$	frequency	pure water	discrete (N = 5)	3.18–4.47	34%	1.41	69–80%
[70]	bow-tie loaded dipole ^e^	yes	130×50×7.5 $\left(0.61\lambda_{0}\right)$	frequency	pure & sea water	discrete	0.9–1.4	44%	1.56	>90%

*^a^* data shown here is for the case of a variable probe length, *^b^* not explicitly demonstrated, *^c^* data shown here is for a two-element antenna, *^d^* data shown here is for the RHCP case, ^e^ the data provided here is for the 100 mm dipole.

**Table 4 materials-13-01863-t004:** Liquid Variations, Liquid Solutions and Other Combinations.

Ref.	Antenna Type	Pla-Nar	Dimensions mm^3^ $\left(\lambda_{0}\right)$	Reconfig. Type	Liquid Type	Tuning Type	Frequency Range (GHz)	TR Equation (Equation 1)	TX Equation (Equation 3)	Radiation Efficiency
[49]	cone	no	100×100×41 $\left(2\lambda_{0}\right)$	frequency	TPC	contin. *^a^*	1.6–5.0	103%	3.125	N/A
[50]	spiral *^b^*	no	100×100×41 $\left(0.93\lambda_{0}\right)$	frequency	Choline L-alanine	contin. *^a^*	a few bands between 1.37 and 2.8	-	-	N/A
[75]	mono- pole	no	30×28.6×28.6 (0.43λ)	frequency	BSTO in oil	contin.	2.75–4.22	42%	1.54	N/A
[76]	Yagi- Uda	no	950×100×80 $\left(0.56\lambda_{0}\right)$	frequency	pure water & sea water	contin.	0.069–0.177	88%	2.57	40–74%
[78]	dual-tube mono- pole	no	39.5×9.5×9.5 $\left(0.37\lambda_{0}\right)$	frequency	pure water & sea water	contin.	1.57–2.83	57%	1.8	52–84%
[80]	mono- pole	no		pattern	glass & ethyl acetate	discrete (N = 2)	3.75–5.37	-	-	>80% 50–63%

*^a^* not explicitly demonstrated in the paper, *^b^* the data shown here is for the case without the metasurface lid.

**Table 5 materials-13-01863-t005:** Planar antennas with microfluidic channels.

Ref.	Antenna Type	Pla-Nar	Dimensions mm^2^ $\left(\lambda_{0}\right)$	Reconfig. Type	Liquid	Tuning Type	Frequency Range (GHz)	TR Equation (Equation 1)	TX Equation (Equation 3)	Radiation Efficiency
[81]	mono- pole	yes	30×40 $\left(0.78\lambda_{0}\right)$	frequency	pure water	discrete (N = 4)	4.44–5.82	27%	1.3	>90.4%
[82]	patch	yes	90×60 $\left(0.56\lambda_{0}\right)$	frequency	pure water	discrete (N = 4)	1.39–1.86	29%	1.34	68.8–86.3%
[83]	patch	yes	3.32×2.45 $\left(0.31\lambda_{0}\right)$	frequency	various *^a^*	discrete (N = 4)	16.8–28.4	51%	1.7	27%
[84]	patch *^b^*	yes	135×135 $\left(0.52\lambda_{0}\right)$	freq., po- larization	pure water	contin.	0.700–1.150	49%	1.64	77–82%
[85]	annular slot	yes	14.8×23.05	frequency	various *^c^*	discrete (N = 6)	3.3–4.2 and 5.2–8	24% 42%	1.27 1.54	73.4% 39.6%
[87]	mono- pole	yes	90×45 $\left(1.05\lambda_{0}\right)$	frequency	FC-40	contin.	1.7–3.5	69%	2.06	87.4–92.4%
[89]	patch	yes	142.1×83 $\left(1.2\lambda_{0}\right)$	frequency	mineral oil	contin.	2.22–2.53 *^d^*	13%	1.14	27–38%
[90]	Vivaldi	yes	250×150 $\left(3.75\lambda_{0}\right)$	frequency	KCl solution	discrete (N = 2)	2.5–3.25 and 3.25–4.50	26% 32%	1.30 1.39	70–87%
[86]	patch	yes	95×95 (0.76λ)	polariza- tion	water & air	discrete (N = 2)	∼2.4	-	-	22%

*^a^* water, acetone, methanol, ethanol, *^b^* the data shown here is for the design III in LP, *^c^* air, water, acetone, *^d^* for the first mode.

**Table 6 materials-13-01863-t006:** Planar antennas with substrate cavities.

Ref.	Antenna Type	Pla-Nar	Dimensions mm^2^ $\left(\lambda_{0}\right)$	Reconfig. Type	Liquid	Tuning Type	Frequency Range (GHz)	TR Equation (Equation 1)	TX Equation (Equation 3)	Radiation Efficiency
[91]	patch	yes	168×168 $\left(1.1\lambda_{0}\right)$	frequency	transform-er oil	contin.	1.42–1.96	31%	1.38	>87%
[92]	spiral	yes	N/A	frequency	air	contin.	0.426–0.542	24%	1.27	2–55%
[93]	patch	yes	10×23 $\left(4.2\lambda_{0}\right)$	frequency	air	contin.	51.00–55.35	8%	1.09	36–39%
[94]	patch	yes	16×16 $\left(0.3\lambda_{0}\right)$	frequency	air	contin.	5.7–6.1	7%	1.07	N/A
[95]	patch	yes	3.85×3.85 $\left(0.3\lambda_{0}\right)$	frequency	silicone oil	contin.	23.28–23.53	1%	1.01	N/A
[96]	patch	yes	30×30 $\left(0.9\lambda_{0}\right)$	frequency	air	contin.	N/A	3.9%	-	32%
[103]	bent dipole *^a^*	no	120×65×40 $\left(0.58\lambda_{0}\right)$	frequency	ethyl acetate	contin.	0.90–1.44	46%	1.6	69–92%

*^a^* data provided here are for the case of the ethyl acetate.

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
