# Peer review of "A Review on Reconfigurable Liquid Dielectric Antennas"

_materials, 2020, doi:10.3390/ma13081863_

Round 1

Reviewer 1 Report

The paper is a good one and deserves publication as a review paper.

The title of the contribution is precise, well-formulated and it fully corresponds the treated topic. The authors have a high level of understanding of current research, explanatory frameworks and problems within the field of inquiry. . The paper is well-structured in Terms of a review paper having a well-written abstract which reflects the content of the article. The authors clearly stated the problem under investigation in the introduction, summarized relative research. The article identifies the procedures followed. The contribution is written with clarity. The narrative is logical and coherent. The authors used the appropriate techniques for analysis of the research objects in order to meet aims of the study. The accurate interpretation of outcomes, well substantiated by the results of the analysis has been achieved by them. The presentation of the results  has been successfully made. The authors have been able to draw logical conclusions from the results. Conclusions are accurate and clearly based on outcomes. 

Please improve the quality of the figures.

Author Response

Dear reviewer,

Thank you for providing your comments! We really appreciate your valuable opinion and time!

As for your suggestion, we have improved the Figures. We noticed that in the pdf version some of the figures were displayed with spurious lines. We have corrected all the figures affected by this error. Moreover, we changed the font size in the labels of Figure 5 to make it more readable.

With kind regards,

the authors 

Reviewer 2 Report

In this paper the authors presented the state-of-the-art techniques and methods used to design, optimize, and apply liquid dielectric reconfigurable antennas. The paper is well written, the main ideas are well presented and in general the content is very interesting. 
However in my opinion the Liquid Dielectric Antennas must be see not only as reconfigurable antenna but also as sensor. It's important to relate the concept of sensor with the use of liquid antennas. Moreover, the manuscript could be improved if news reconfigurable solutions are presented but in relation to the radiation pattern and polarization. Finally the reference to some antenna array it will be important. 

Author Response

Dear reviewer,

Thank you for providing your comments! We really appreciate your valuable opinion and time!

Regarding your questions and comments, kindly see the answers below.

Q1. However in my opinion the Liquid Dielectric Antennas must be see not only as reconfigurable antenna but also as sensor. It's important to relate the concept of sensor with the use of liquid antennas.

A. We understand the reviewer’s concern. Indeed, many of the reconfiguration ideas presented here can be applied to sensor design as the changing environmental conditions will alter the electromagnetic response of a liquid dielectric based antenna and thus can be used for sensing and provide a useful information. Moreover, these reconfiguration concepts can be also applied to the design of filters, arrays, frequency selective surfaces, and metasurfaces. However, all these topics fall beyond the scope of this review for a number of reasons. Firstly, rather than to disperse and try to capture all possible designs, we decided to focus on a specific domain of active antennas and thoroughly review it. Secondly, another goal of this review is to establish a classification of reconfiguration techniques which can help to better understand what has been done and what are the research perspectives, gaps, and limitations. And finally, the concepts highlighted in this review can be brought and applied to other fields of electromagnetic research without losing the generality. To explain and justify our choice for the review material, we added extra sentences to the Introduction. Kindly refer to page 3, lines 97-103 of the revised manuscript.

Q2. Moreover, the manuscript could be improved if news reconfigurable solutions are presented but in relation to the radiation pattern and polarization. Finally the reference to some antenna array it will be important.

A2. We covered reconfiguration solutions that tune the working frequency of the antennas and affect the radiation pattern and polarization. The frequency tuning ranges are summarized in the tables and their relations with the change of the radiation pattern and polarization are discussed in the content. For each type of the reconfiguration (frequency, patter, or polarization), the corresponding information is provided in the fifth column of each table (Kindly refer to Tables 3-5 for the examples of pattern and polarization reconfigurable antennas). As presented in the Discussion section, the majority of the research work in the literature is focused on frequency reconfiguration, with around 74% of all reviewed work focused on frequency reconfiguration, 18% - on pattern reconfiguration, and only 8% - on polarization reconfiguration.

With kind regards,
the authors 

Reviewer 3 Report

Thanks for providing a comprehensive review on reconfigurable liquid dielectric antennas. My comments on this work are as follows:

(1) Figure 1: Since its comprehensive review need to be referenced each part of the flowchart with even multiple references.

(2) For electrically reconfigurable antennas, there are many more types instead of the shown one. Please include it. Some recommendations are: (a) Electronics 20198(8), 900 (b) Sensors 201818(9), 3155.

(3) " Modern wireless communication applications require an antenna to have multiple functionalities  (e.g. beam steering, direction finding, radar, control, and command) within a limited space [1]." Please explain in detail in light of the following references: (a) A Systematic Methodology for the Time-Domain Ringing Reduction in UWB Band-Notched Antennas (b) Isolation Enhancement of Wide-Band MIMO Array Antennas Utilizing Resistive Loading (c) Electronics 2019, 8(2), 158, (d) Electronics 2020, 9(1), 71

(4) Equations need to be referenced from literature with basic ones.

(5) Please provide some advantages of DRA antennas in the table and in a brief way.

(6) Feeding techniques of liquid antennas are well explained.

(7) Please provide advantages and drawbacks of various geometries that have been reviewed in term of the comprehensive table.

(8) Please explain and provide further information regarding the tuning of liquid antennas.

(9) finally, the topic may be modified as: Reconfigurable and tunable liquid antennas

(10) By adding tunable mechanism and comprehensive review in terms of radiation efficiency and other parameters, would increase the scope of the proposed review.

Thank you

Author Response

Dear reviewer,

Thank you for providing your comments! We really appreciate your valuable opinion and time!

Regarding your questions and comments, kindly find the answers below.

Q1. Figure 1: Since its comprehensive review need to be referenced each part of the flowchart with even multiple references.

A1. The flowchart presented in Figure 1 is for the general antenna reconfiguration techniques, which is not the intended focus of the current review. This flowchart is based on the classification of antenna reconfiguration techniques proposed in reference [4]. In [4], the authors provided a comprehensive review on general antenna reconfiguration strategies. That is why we did not provide specific references on the flowchart itself, but rather navigate a reader in the field of reconfiguration techniques. However, we included the reference [4] in the caption of the flowchart (see Figure 1 on page 2) to indicate the relevance to the original source.

On the other hand, we agree with the reviewer that each part of the current review should include specific references. Therefore, we have added the corresponding references for each schematic illustration figures. Kindly refer to the changes in the following captions to Figures 2-6.

Q2. For electrically reconfigurable antennas, there are many more types instead of the shown one. Please include it. Some recommendations are: (a) Electronics 2019, 8(8), 900 (b) Sensors 2018, 18(9), 3155.

A1. Thank you for the recommendations! We have read the recommended papers carefully and found that (a), which offers an electrically driven reconfiguration technique, is very relevant to our manuscript. It is added in Introduction Section at page 2, lines 50-51. For the other paper, as it is less relevant, we are not going to add it for this current manuscript. Hope to have the reviewers’ understanding.

Q3. " Modern wireless communication applications require an antenna to have multiple functionalities  (e.g. beam steering, direction finding, radar, control, and command) within a limited space [1]." Please explain in detail in light of the following references: (a) A Systematic Methodology for the Time-Domain Ringing Reduction in UWB Band-Notched Antennas (b) Isolation Enhancement of Wide-Band MIMO Array Antennas Utilizing Resistive Loading (c) Electronics 2019, 8(2), 158, (d) Electronics 2020, 9(1), 71

A3. Thank you for the recommendations! We have read the recommended papers very carefully. However, we would like to emphasise that the purpose of this review is to systematically analyse liquid dielectric based reconfigurable antennas. For this reason, electrically reconfigurable antennas are covered only briefly in this work. Therefore, we could only include one reference (d) Electronics 2020, 9(1), 71, which offers a novel MIMO antenna array. Kindly refer to page 1, line 19. Hope to have the reviewers’ understanding.

Q4. Equations need to be referenced from literature with basic ones.

A4. The corresponding references were provided for each equation. Kindly refer to page 3, lines 106, 110, page 4, line 114, and page 5, line 193, page 18, line 570.

Q5. Please provide some advantages of DRA antennas in the table and in a brief way.

A5. The basic liquid DRA designs are summarised in Table 1 together with a brief analysis. Kindly refer to page 6, lines 217-218.

Q6. Feeding techniques of liquid antennas are well explained.

Q6. Thank you!

Q7. Please provide advantages and drawbacks of various geometries that have been reviewed in term of the comprehensive table.

A7. The various geometries for liquid dielectric reconfigurable antennas are summarized in Table 3. Kindly refer to page 11.

Q8. Please explain and provide further information regarding the tuning of liquid antennas.

A8. For each antenna design discussed in this manuscript its tuning technique is specified in the seventh column of the corresponding table. Moreover, the tuning range TR, the type of tuning (continuous or discrete) and the tuning material are all summarised in each table. Kindly refer to Tables 1-6.

Q9. finally, the topic may be modified as: Reconfigurable and tunable liquid antennas

A9. Thank you for the suggestion! We respect the reviewer’s opinion, but we believe it is the matter of preferences. Therefore, we prefer to stick to the title of our choosing.

Q10. By adding tunable mechanism and comprehensive review in terms of radiation efficiency and other parameters, would increase the scope of the proposed review.

A10. All the tuning mechanisms discussed in this work rely on the change of liquid volume, liquid height, or the type of liquid dielectric, as discussed on pages 18-19, and the introduction to Section 3, on page 13. The radiation efficiency is an important parameter and it is included for each discussed antenna in the last column of each summary table. Kindly refer to Tables 1-6.

We hope the provided answers are sufficient for the reviewer. Thank you! 

With kind regards,
the authors 

Reviewer 4 Report

Dear authors,

This paper is a good review of liquid dielectric antennas. The paper is very well structured and the references are very well explained and compared among them. 

Only two minor comments:

  1. In the abstract it is mentioned: "the required DC circuitry can significantly decrease the radiation efficiency of such antennas". In my opinion, this is not general and it depends on the kind of switching elements. 
  2. A typo error: in line 38: it is written THey instead of They.

Sincerely,

Author Response

Dear reviewer,

Thank you for providing your comments! We really appreciate your valuable opinion and time!

Regarding your comments, kindly find the answers below.

Q1. In the abstract it is mentioned: "the required DC circuitry can significantly decrease the radiation efficiency of such antennas". In my opinion, this is not general and it depends on the kind of switching elements.

A1. Thank you for pointing this out! We modified the sentence to be more general. Please find the modified sentence on page 1, lines 7-8.

Q2. A typo error: in line 38: it is written THey instead of They.

A2. Thank you for pointing this out! The typo was corrected.

With kind regards,
the authors 

Round 2

Reviewer 2 Report

In this new version of the manuscript, I think some important aspects were clarified. Moreover the answers of the authors are acceptable and for this reason I recommend the publication of the manuscript.

Reviewer 3 Report

Thanks for your response.

I believe that this review got possible information for readers to publish at the current stage.

No further concerns.